# *Corydalis saxicola* Bunting: A Review of Its Traditional Uses, Phytochemistry, Pharmacology, and Clinical Applications

**DOI:** 10.3390/ijms24021626

**Published:** 2023-01-13

**Authors:** Feng Qin, Yao Chen, Fan-Fan Wang, Shao-Qing Tang, Yi-Lin Fang

**Affiliations:** 1State Key Laboratory for Chemistry and Molecular Engineering of Medicinal Resources, Collaborative Innovation Center for Guangxi Ethnic Medicine, School of Chemistry and Pharmaceutical Sciences, Guangxi Normal University, Guilin 541004, China; 2Ministry of Education Key Laboratory for Ecology of Rare and Endangered Species and Environmental Protection, School of Life Sciences, Guangxi Normal University, Guilin 541004, China

**Keywords:** botany, *Corydalis saxicola*, clinical applications, phytochemistry, pharmacology, traditional uses

## Abstract

*Corydalis saxicola* Bunting (CSB), whose common name in Chinese is Yanhuanglian, is a herb in the family Papaveraceae. When applied in traditional Chinese medicine, it is used to treat various diseases including hepatitis, abdominal pain, and bleeding haemorrhoids. In addition, *Corydalis saxicola* Bunting injection (CSBI) is widely used against acute and chronic hepatitis. This review aims to provide up-to-date information on the botanical distribution, description, traditional uses, phytochemistry, pharmacology, and clinical applications of CSB. A comprehensive review was implemented on studies about CSB from several scientific databases, such as SciFinder, Elsevier, Springer, ACS Publications, Baidu Scholar, CNKI, and Wanfang Data. Phytochemical studies showed that 81 chemical constituents have been isolated and identified from CSB, most of which are alkaloids. This situation indicates that these alkaloids would be the main bioactive substances and that they have antitumour, liver protective, antiviral, and antibacterial pharmacological activities. CSBI can not only treat hepatitis and liver cancer but can also be used in combination with other drugs. However, the relationships between the traditional uses and modern pharmacological actions, the action mechanisms, quality standards, and the material basis need to be implemented in the future. Moreover, the pharmacokinetics of CSBI in vivo and the toxicology should be further investigated.

## 1. Introduction

*Corydalis saxicola* Bunting (CSB), which is called Yanhuanglian in Chinese, is a species of *Corydalis* DC., is a genus in the family Papaveraceae. CSB is mainly distributed on rocky cliffs or in alpine caves in southwestern China, including Guangxi, Guizhou, Hubei, Shanxi, Sichuan, Yunnan, Zhejiang [1]. CSB is a traditional Chinese folk medicine in southwestern China, where the bitter-tasting and cool-natured whole plant can be used as a medicine that has the effects of heat clearing and detoxification, damp elimination, pain alleviation, and haemostasis [2]. In tradition, the whole plant of CSB, namely Yan-huang-lian is used to cure diseases in Chinese folk. On the basis of traditional Chinese medicine theories, its use spectrum contains acute conjunctivitis, corneal pannus, acute abdominal pain, hemorrhoidal bleeding, haematochezia, swelling, hepatitis, cirrhosis, and liver cancer [3,4,5]. Until now, pharmacological studies have provided evidence that this ethnomedicine can treat liver diseases and show that it also has significant pharmacological properties such as anticancer, anti-inflammatory, antibacterial, anti-oxidative, analgesic, and hepatoprotective effects [6,7,8]. Clinically, *Corydalis saxicola* Bunting injection (CSBI) was approved for production in Guangxi in 1982 [1] and is the exclusive variety in production in China. In 2002, the national medicine standard (Z20026725) was obtained by landmark upgrading and national standard rectification. CSBI is clinically used for patients with acute and chronic hepatitis.

Although a large number of studies have been carried out on CSB, many aspects of CSB remain unclear, such as monomer component exploration, the establishment of quality standards, the mechanism of CSB therapeutic effects, and combination mechanism. Additionally, toxicological studies need to be strengthened to support its therapeutic safety. In this paper, the botanical properties, traditional uses, phytochemistry, pharmacological activities, and clinical applications of CSB are reviewed. This review can provide a reference for deep research and application of this plant.

## 2. Database Search Method

This review of CBS on its botanical distribution and description, traditional uses, phytochemistry, pharmacological activity, clinical applications, and toxicity assessment is based on a variety of reliable databases such as Scifinder, PubMed, ScienceDirect, Wiley, ACS, CNKI, Springer, Google Scholar, Web of Science, and Baidu Scholar, as well as other published materials (Ph.D. and M.Sc. dissertations and books). The literature was searched and accessed using the keywords “*Corydalis saxicola* Bunting”, “*Corydalis saxicola* Bunting total alkaloids”, and “dehydrocavidine” which are related to the present review.

## 3. Botanical Distribution and Description

CSB can be found in southwestern China with a wide distribution in Chongqing (Chengkou), Guangxi (Debao, Fengshan, Jingxi), Guizhou (Dushan, Weng’an, Zunyi), Hubei (Yichang), Shanxi (Mianxian), Sichuan, Yunnan (Xichou), and Zhejiang (Ningbo), among other areas [9,10]. Notably, it is mainly distributed in western and northwestern Guangxi, especially Donglan, Bama, Du’an, Jingxi, and Debao. It has stringent requirements for environmental conditions and is produced in extremely low yield. The distribution of this plant is limited to limestone mountain areas and is a stone mountain endemic species. It grows primarily on rocky cliffs or in alpine caves at elevations of 600–1690 m, reaching 2800–3900 m in southwestern Sichuan, with a growth temperature of 15–25 °C [10,11,12].

CSB is a light green soft herb that can grow up to 30–40 cm high and has thick main roots and single to multi-headed rhizomes. The stems can be branched or unbranched. The branches are opposite to the leaves and are scape-shaped. The basal leaves are approximately 10–15 cm long with a long stalk, the leaves are approximately the same length as the petioles with two or one pinnate cleft, and the last pinnate is wedge-shaped to obovate, approximately 2–4 cm long and 2–3 cm wide, with unequal 2–3 splits or thick round teeth at the edges. The raceme is approximately 7–15 cm long, with many flowers that are first dense, then alienated. The bracts are elliptic to lanceolate, whole, and the lower part is approximately 1.5 cm long and 1 cm wide while the upper part narrows, all of which are longer than the pedicels. The length of the pedicel is approximately 5 mm. The flowers are golden yellow and flat. The sepals are like triangles, and in their entirety are approximately 2 mm long. The outer petals are wider and acuminate, and the coronal process is limited to the keel process but does not reach the top. The upper petals are approximately 2.5 cm long with the distance accounting for approximately 1/4 of the full length of the petals, slightly bent down, and the end is cystic. The lower petals are approximately 1.8 cm long, and the base looks like a small tumour protrusion. The inner petals are approximately 1.5 cm long, with a thick crown protruding from the top. The stamen bundles are lanceolate, tapering above the middle. The stigma is two-forked, with two cleft papillae at the top of each branch. The capsule is linear and recurved, approximately 2.5 cm long, with 1 row of seeds [10]. (see Figure 1).

## 4. Traditional Uses

As a type of folk medicinal herb in South China, various classical ancient Chinese books, including “Guizhou folk medicine”, “Traditional Chinese Medicine Dictionary”, “Chinese Materia Medica”, “Flora of Guangxi”, and “Flora of Yunnan”, have recorded the traditional medicinal uses of CSB. Based on the theory of traditional Chinese medicine, CSB has been widely used in the treatment of various diseases for a long time, such as heat clearance and detoxification, damp elimination, pain alleviation, haemostasis, erosion of the mucous membranes in the oral cavity, dysentery, eyes of fire, eye shade, diarrhoea, stomachache, bleeding haemorrhoids, hepatitis, jaundice, ascites, cirrhosis, and hepatoma [13,14,15,16]. For example, direct oral administration of 6 g of CSB is used to treat acute abdominal pain. For the treatment of pile ale bleeding and red dysentery, 15 g of CSB soaked in 50 g of steamed liquor is taken orally. In addition, CSB can be mixed with other medicines to treat diseases. For example, for the treatment of eyes of fire and eye shade, CSB (3 g), *Gentiana scabra* Bunge (3 g), and “Shang Mei Pian” (1.5 g) are ground together into powder, steamed in a porcelain cup, and finally dipped into the eye with a lantern. An injection containing the total alkaloids of CSB has a good effect on hepatitis, cirrhosis, and hepatoma [12,17]. CSB is mainly used orally (decoction, 3–15 g) and externally (put on the affected area after grinding in the appropriate amount). When using CSB, dry and spicy foods should be avoided. Due to diverse traditional and folk uses of it, CSB has been extensively studied by numerous chemists and biologists.

## 5. Phytochemistry

A research team from Guangxi in China reported on CSB in 1980. To date, 81 chemical components including alkaloids, steroids, and flavonoids, have been separated and identified from CSB (Table 1). The chemical structures of these 81 compounds are shown in Figure 2.

### 5.1. Alkaloids

Alkaloids are key secondary metabolites of CSB. Due to the diversity of their structures and biological activities, an increasing number of studies on alkaloids have been conducted in China in recent decades. To date, 67 kinds of alkaloids have been identified and reported in CSB, such as berberines, protoberberines, benzophenanthridines, protoberberines, benzyltetrahydroisoquinolines, aporphines, protopines, and lignin amides (**1**–**67**) (Table 1). Six alkaloids with high purity were obtained from the crude n-butanol extract of CSB by high-speed counter-current chromatography: (−)-scoulerine (**30**), (+)-isocorydine (**50**), dehydrocheilanthifoline (**8**), dehydrocavidine (**1**), palmatine (**9**), and berberine (**4**) [24]. To investigate the phytochemical composition of CSB, the whole plant is usually selected as the source. As a rapid separation strategy, high-speed counter-current chromatography is the major majority approach to the isolation of the main components of it. The results basically show that alkaloids are the main components of CSB and the content of dehydrocavidine is the highest one.

### 5.2. Others

With the exception of alkaloids, 14 non-alkaloids were isolated and identified from CSB, including 9 steroids (**68**–**76**), 2 flavonoids (**77**–**78**), 2 alkanes (**79**–**80**), and 1 uracil (**81**) [22,23]. The research on other types of natural products is rarely performed apart from the alkaloids. Therefore, it is urgent to explore novel leading compound with biological activity in future discovery.

## 6. Pharmacology

In recent decades, modern pharmacological studies have shown that CSB has a wide range of pharmacological activities, such as anticancer, hepatoprotective, anti-hepatitis B virus (HBV), central nervous system, immune function enhancement, anti-inflammatory, analgesic, antioxidant, antibacterial, and choleretic effects. A large number of studies have reported that the crude extracts and chemical constituents of CSB have strong biological activity both in vivo and in vitro. In addition, research on the activity of CSB has mainly focused on alkaloids. The pharmacological effects of CSB reported in recent years are summarized in Table 2.

### 6.1. Anticancer Activity

#### 6.1.1. Crude Extracts

Cancer is still the leading cause of mortality among humans. Many parts of the body can trigger the growth of cancer cells, and the emergence of cancer poses a very large threat to human health. Recent studies have provided evidence that the anticancer activity of CSB is mainly related to its alkaloid compounds. *Corydalis saxicola* Bunting total alkaloids (CSBTA) has shown cytotoxicity against five cancer cell lines: CNE-1 (IC_50_ = 112.41 µg/mL), CNE-2 (IC_50_ = 123.46 µg/mL), A2780 (IC_50_ = 148.40 µg/mL), SKOV3 (IC_50_ = 128.51 µg/mL), and PM2 (IC_50_ = 166.66 µg/mL) [23].

CSB extracts can increase actin filament disassembly and cofilin-1 activity, and decrease Cdc42 protein expression in A549 cells to reduce the migration ability of cells [27]. Additionally, CSBTA inhibits migration of A549 cells by suppressing cell division cycle 42 or vav guanine nucleotide exchange factor 1 [8]. Another study found that the effects of CSBTA on the proliferation of A549 lung cancer cells displayed a time–dose relationship; with increasing concentration of CSBTA, the apoptosis rate of A549 cells increased, the expression of caspase-3 mRNA was upregulated, and the expression of survivin mRNA was downregulated [28]. Moreover, CSBTA combined with cisplatin can enhance the inhibition of proliferation of A549 cells [29].

A study confirmed that CSBTA inhibited the growth of A549 cell-transplanted tumours by downregulating survivin in nude mice, and it can also reduce the degree of bone destruction in bone metastases [26]. Furthermore, CSBTA attenuate Walker 256-induced bone pain and osteoporosis by suppressing the RANKL-induced NF-κB and c-Fos/NFATc1 pathways in rats [25].

The above five studies reported the activity of CSBTA on A549 cells, while there have also been studies on CSBTA in Tca8113 cells. CSBTA can significantly inhibit the proliferation and apoptosis of human tongue squamous cell carcinoma Tca8113 cells in a time-dependent and dose-dependent manner [31]. Additionally, [30] found that CSBTA could promote apoptosis of Tca8113 cells by inhibiting the expression of Bcl-2 at the mRNA and protein levels. [32]. Overall, as a leading compound targeting tumour pathway, CSBTA is promising for developing into a novel anticancer drug.

CSBI has been reported to exert a certain inhibitory effect on mouse S180 sarcoma and Ehrlich ascites tumours, with an average inhibition rate of more than 30%. In addition, the combination of CSBI and phagocytic acid can enhance the antitumour effects, with the average tumour inhibition rate reaching over 50% [33]. The antitumour activity of CSBTA was further evaluated through in vitro, semi-in vivo, in vivo, methylene blue reduction, and cell staining methods, and the results showed that CSBTA had a certain inhibitory effect on sarcoma 180 Ehrlich ascites carcinoma in mice and sarcoma 256 in rats [17,34,35]. Lu’s test reported that CSBTA could inhibit respiratory metabolism in S180, HAC, and EAC tumour cells [34]. The water extract of CSB can inhibit the proliferation and migration of HepG2 liver cancer cells, and its mechanism may be related to the upregulation of NF-κB p65 expression [25]. Therefore, the water extract of CSB and CSBTA has excellent antitumour activity against a variety of tumour cells. However, what specific components take effect in the water extract is still unclear and needs to be further explored.

#### 6.1.2. Isolated Phytochemicals

Eight lignanamides (corydalisin A (**58**), corydalisin B (**59**), corydalisin C (**61**), cannabisin F (**60**), cannabisin E (**64**), cannabisin D (**62**), 1,2-dihydro-6,8-dimethoxy-7-hydroxy-1-(3,5-dimethoxy-4-hydroxyphenyl)-N1,N2-bis-[2-(4-hydroxyphenyl)ethyl]-2,3-naphthalene dicarboxamide (**63**), and grossamide (**65**)) exhibited anticancer activity against five tumour cell lines (MGC-803, HepG2, T24, NCI-H460, and Spca-2; IC_50_ > 8.81 ± 2.05 μM). Meanwhile, corydalisin C may induce apoptosis via both the intrinsic and extrinsic apoptosis pathways and downregulation of Bcl-2 and FasL expression in a time-dependent manner [1]. Dehydrocavidine (**1**) and palmatine (**9**) can inhibit the proliferation of hepatocellular carcinoma SMMC-7721 cells [37]. Dehydrocavidine (**1**) significantly inhibited the proliferation of human tongue squamous cell carcinoma Tca8113 cells and decreased the expression of NF-κB, telomerase activity and the expression of hTERT [38,39]. (−)-Pallidine (**51**) and (−)-scoulerine (**30**) have strong inhibitory effects on human DNA topoisomerase I [19]. Chelerythrine (**15**) and (−)-13β-hydroxystylopine (**23**) have certain anticancer effects [21]. Corysaxicolaine A (67) displayed obvious inhibitory effects on the tested human cancer cells (T24, A549, HepG2, MGC-803, SKOV3), with IC_50_ values of 7.63, 13.32, 12.39, 9.98, and 12.36 μM, respectively [7]. However, there is little research on the mechanisms of these individual compounds or their anticancer activity in vivo. Elucidating the anticancer activity and mechanism of CSBTA and its individual compounds should be the focus of future research.

### 6.2. Hepatoprotective Effects

#### 6.2.1. Crude Extracts

A large number of studies have reported that CSB plays a pivotal role in the treatment of acute and chronic liver injury, liver fibrosis, and liver cirrhosis. Some researchers found that the expression levels of ALT, AST, MDA, and ALP in serum were significantly decreased, and ALB, SOD, and GSH was significantly increased in liver tissue of rats with acute and chronic liver injury after CSBTA intervention [20,39,40,41,45,46,48,49,50,51]. Moreover, CSBTA has an obvious protective effect on acute liver injury caused by CCl_4_ and can also inhibit the formation of chronic liver fibrosis in rats. The potential mechanisms for this protective effect may be related to anti-lipid peroxidation, inhibition of collagen synthesis, and the promotion of extracellular matrix degradation [48]. However, Bi [46] showed that 0.78 mg/kg and 2.34 mg/kg CSBTA could significantly reduce hepatocyte degeneration, necrosis, inflammatory cell infiltration, and the activity of serum ALT but had no obvious effect on the activity of AST.

Some studies found that CSBTA has hepatoprotective and anti-fibrotic effects on rats with chronic liver fibrosis [47], which may promote the reversal of liver fibrosis by inhibiting the expression of TGF-β1 and MMP-9 [49]. The effects of CSBTA on chronic liver injury induced by CCl_4_ may include the regulation of tricarboxylic acid (TCA) circulation, intestinal microbial metabolism, and taurine and hypotaurine metabolism disorder [43]. Furthermore, researchers reported that dehydrocavidine (**1**), palmatine (**9**), and berberine (**4**) were shown to induce apoptosis and autophagy in HSC-T6 cells, and suggested that these three components may be the components in CSBTA that are effective against liver fibrosis [41]. Through the biotransformation mediation of CSBTA by CYP450s in the liver, CSBTA-drug interactions might occur through CYP450s inhibition, particularly CYP1A and CYP2D. [42].

This CSB extract could significantly reduce the degree of liver damage, exert a protective effect on acute liver injury caused by acetaminophen [54] and CCl_4_ [53], respectively. Meanwhile, the therapeutic effects of this CSB extract on acute liver injury induced by CCl_4_ may be related to regulation of alanine, aspartic acid, glutamic acid, and glycerol metabolic dysfunction [52,55].

Betulinic acid (**71**), β-amyrin acetate (**73**), and (−)-pallidine (**51**) in a CSB extract may play an anti-fibrotic role by regulating the targets FXR, COX-2, and MMP-1 [78]. ^1^H NMR was used to study the changes in the sera of rats with liver fibrosis induced by CCl_4_ treated with this CSB extract, and partial least squares discriminant analysis showed that metabolic disorders were reduced after CSB treatment [51]. In summary, the protective efficiency and anti-fibrosis effects on acute and chronic liver injury of CSBTA provide the scientific basis for the utilization of CSB resources. However, the discrepancy of action mechanism between monomer compounds and extracts needs further explanation.

#### 6.2.2. Isolated Phytochemicals

Through the study of group–effect relationships, Lu [56] reported for the first time that dehydrocavidine (**1**), palmatine (**9**), and berberine (**4**) could significantly inhibit the proliferation of HSC-T6 cells and induce their apoptosis in vitro without obvious cytotoxic effects at effective concentrations. Dehydrocavidine (**1**) can significantly reduce ALT, AST, and total bilirubin (TBIL) in mice with acute liver injury and effectively reduce CCl_4_-induced hepatocyte degeneration, necrosis, inflammatory cell infiltration, and ultrastructural destruction [59]. In a rat model of liver fibrosis induced by CCl_4_, dehydrocavidine (**1**) could reduce the degree of liver injury and the formation of interstitial fibrous tissue, and its mechanism of liver protection and liver fibrosis inhibition may be related to the extracellular matrix and antioxidant stress [57,58]. To date, most studies have focused on the effects of CSBTA on liver injury and fibrosis, while few studies have focused on its contained individual compounds. More studies should be undertaken that focus on this field.

### 6.3. Anti-HBV Activity

#### 6.3.1. Crude Extract

HBV causes the disease hepatitis B, and it is also infectious to a certain extent. CSB has been reported to have very good anti-HBV activity. Wang’s experiment proved that the extract of CSB has anti-duck hepatitis B virus (DHBV) effects in vivo, as the level of serum DHBV DNA in all CSB groups decreased significantly after treatment. Pathological examination showed that the CSB extract had a protective effect on liver injury induced by DHBV [60].

#### 6.3.2. Isolated Phytochemicals

Zeng et al. [3] isolated dehydrocheilanthifoline (**8**) from CSB and showed its anti-HBV activity for the first time. The inhibition of HBsAg (IC_50_ 17.12 μM) and HBeAg (IC_50_ 15.58 μM) by dehydrocheilanthifoline was dose- and time-dependent, with obvious anti-HBV activity in vitro. Moreover, the anti-HBV activities of certain compounds isolated from CSB were evaluated, and dihydrochelerythrine (**18**) and (−)-salutaridine **(54**) were found to have strong inhibitory effects on HBsAg and HBeAg with IC_50_ values of <0.02, <0.02 mg/mL for dihydrochelerythrine (**18**), respectively, and 0.09, 0.15 mg/mL for (−)-salutaridine (**54**), respectively [61]. Dehydrocavidine (**1**), dehydroapocavidine (**2**), and dehydroisoapocavidine (**3**) had obvious inhibitory effects on HBsAg and HBeAg and showed no toxicity in 2.2.15 cells [15]. Furthermore, the anti-HBV activities of 10 major alkaloids were tested. Among the tested compounds, dihydrochelerythrine (**18**) exhibited the most potent activity against HBsAg and HBeAg secretion, with IC_50_ values < 0.05 μM and selectivity index (SI) values > 3.5 [18]. The research mentioned above showed the application value in the prevention and treatment of HBV of CSB. The practical values related to prevention and treatment of HBV indicate that the prevention of HBV utilizing dihydrochelerythrine (**18**) demands further studies on its mechanism of action.

### 6.4. Enhancement of Immune Function

#### Crude Extract

Immune function is the body’s resistance to diseases. The immune function of the human body is mainly manifested in three aspects: immunoligic defence, immunoligic homeostasis, and immunoligic surveillance. A study showed that CSBTA enhanced the haemolytic plaque value and delayed-type hypersensitivity in mice in vivo, enhanced the mixed culture response of allotypic mice splenocytes in vitro, and enhanced the proliferation response of splenocytes stimulated by mitogen. In addition, CSBTA increased the levels of IFN-γ and IL-2 produced by T cells in a dose-dependent manner, indicating that CSBTA is an enhancer of immune regulation [62]. It can be seen from the literature results that CSBTA has a certain effect on enhancing immune function.

### 6.5. Antioxidant Activity

#### 6.5.1. Crude Extract

The IC_50_ refers to the measured semi-inhibitory concentration of an antagonist 2,2-Diphenyl-1-picrylhydrazyl radical (DPPH) is a relatively stable free radical and which is frequently used as a reactivity model of reactive oxygen species. In a study, when the DPPH radical was eliminated, the absorbance A at the maximum absorption wavelength of 519 nm decreased. CSBTA enhanced the antioxidative ability of rat livers by reducing the content of MDA and increasing the activity of SOD in a dose-dependent manner, which indicated that CSBTA can directly or indirectly alleviate liver cell injury and inflammation through antioxidation, thus playing a very good role in protecting the liver [63].

#### 6.5.2. Isolated Phytochemicals

The IC_50_ of the free radical scavenging activity of nine alkaloids derived from CSB in vitro ranged from 0.25 to 16.51 mg/mL [2]. The structure–activity relationship study of the antioxidant activity showed that the antioxidant activities of compounds with basic tertiary amines were higher than those of their corresponding basic quaternary amine compounds [2]. The kinetics results showed that the compounds with the highest activity, (−)-pallidine (**51**) and (+)-cheilanthifoline (**36**), had non-linear DPPH radical scavenging activity, while the other alkaloids displayed linear DPPH radical scavenging activity, indicating that their DPPH radical scavenging activity was dose-dependent [2]. Despite its strong antioxidant activity, many unknown mechanisms of alkaloids exist, which demands further investigation.

### 6.6. Effects on the Central Nervous System

#### 6.6.1. Crude Extract

The central nervous system is a group of neurons that regulate a specific physiological function, such as a respiratory center, a thermoregulation center, or a language center. There is increasing evidence showing how phytochemicals influence the central nervous system. They may take effect as anticonvulsants, antidepressants, hypnotic sedation, and anti-senile dementia, and have other pharmacological effects. Some studies have reported that CSB has effects on the central nervous system. A study demonstrated that CSBTA (50, 100 mg/kg) significantly decreased the contents of 3,4-dihydroxyphenylacetic acid (DOPAC), homovanillic acid (HVA), 5-hydroxytryptamine (5-HT), and 5-hydroxyinolacetic acid (5-HIAA) but had no significant effect on the level of dopamine (DA). However, the DA/DOPAC and DA/HVA ratios increased, and the 5-HT/5-HIAA ratio of the limbic system also increased. Thus, it has been suggested that CSBTA can inhibit the metabolism of DA and 5-HT in certain brain regions [64]. Furthermore, CSBTA significantly inhibited caffeine-induced excitatory activity in mice. In general, CSBTA had a calming effect on monkeys and cats. Additionally, in some animals, CSBTA produced catalepsy. The irritation response induced by electrical stimulation was found to be significantly inhibited in mice. The conditioned response of the rats was blocked, but CSBTA showed little effect on the unconditioned response. In a study of the central inhibition mechanism, it was found that CSBTA decreased the levels of 5-HT and 5-HIAA in the striatum [66].

#### 6.6.2. Isolated Phytochemicals

Dehydrocavidine (**1**), isolated from CSB, can increase the contents of DA and 3-dihydroxyphenylacetic acid in the striatum of normal rats and decrease the contents of DA and HVA in the striatum of model rats [67]. The results suggested that dehydrocavidine (**1**) can significantly block central DA function in rats. In short, dehydrocavidine (**1**) is the major contributor of CSB’s effects on the central nervous system.

### 6.7. Anti-Inflammatory Activity

#### Crude Extracts

Inflammation is a body’s protective response to potentially harmful stimuli. The stimuli can initiate the immune system to provide protection, while the protective mechanism depends on neutrophils, macrophages, dendritic cells, and monocytes. Those cells produce various inflammatory mediators (IL-1*β*, IL-6, TNF-*α*, IL-8, chemokines, eicosanoids, histamine, etc.), which results in acute inflammation. Notably, most Chinese herbal medicines have anti-inflammatory properties. A low dose of CSBTA (0.4375 mg/kg) could significantly inhibit the formation of cotton ball granuloma in mice and reduce the chronic inflammatory response [70]. Furthermore, intraperitoneal injection of 50 mg/kg CSBTA could reduce the degree of foot swelling in rats with Danqing arthritis, but the same dose of subcutaneous radiation had no effect on formalin arthritis in rats [66]. CSBTA ameliorates diet-induced non-alcoholic steatohepatitis by regulating hepatic PI3K/Akt and TLR4/NF-κB pathways in mice [4]. It has been reported that *C. saxicola* rectal suppository had obvious inhibitory effects on ear swelling induced by croton oil, on the increase in capillary permeability induced by acetic acid, and on the writhing reaction induced by peritoneal injection of acetic acid [72]. In addition, CSBTA also was shown to effectively suppress M1 polarization of THP-1-derived Mφs, which may improve the inflammatory environment [69]. Some studies have shown that *C. saxicola* suppository has a certain therapeutic effect on rats with chronic pelvic inflammatory disease, shows good anti-inflammatory effects, and regulates immunity. Its mechanism may be related to the regulation of the expression of the inflammatory factors TNF-α and IL-6 and the immune proteins IgG and IgM [71]. Future research should identify which chemical constituents of CSB are responsible for these anti-inflammatory effects.

### 6.8. Analgesic Effect

#### 6.8.1. Crude Extract

Pain is a kind of disease with complex pathogenesis and serious harm to human physical and mental health. Traditional medicines have long been used in the treatment of pain, and numerous medicinal herbs have been reported to be able to relieve pain effectively. Two studies have proven that CSBTA has good analgesic effects. Subcutaneous injection of CSBTA (50 mg/kg, 100 mg/kg) had an obvious inhibitory effect in the writhing reaction of mice, and CSBTA (10 mg/kg) also improved the pain threshold to heat stimulation in rats in the tail flicking test [66]. In another report, Kuai et al. [73] believed that the therapeutic effects of CSBTA were a result of blocking the activation of TRPV1 by improving neuronal damage, improving the loss of intraepidermal nerve fibers (IENFs), and inhibiting inflammation-induced p38 phosphorylation. Therefore, the analgesic effect of CSBTA initially reported on chemotherapy-induced peripheral neuropathy has proposed a novel strategy for the clinical treatment of this disease.

#### 6.8.2. Isolated Phytochemicals

CSB is a Chinese herbal medicine with anti-inflammatory and analgesic effects. The main component of CSB, dehydrocavidine (**1**), has shown sedative and analgesic effects [21,68]. In the future, the chemical composition of the components that produce analgesia need to be further explored and identified.

### 6.9. Antibacterial Activity

#### 6.9.1. Crude Extract

Infections caused by Gram-positive and Gram-negative bacteria are one of the foremost causes of morbidity and mortality globally. However, natural products are the main sources of antimicrobials used in clinical practice, serving as a rich reservoir for the discovery of new antibiotics. As two good indicators for evaluating antimicrobial agents, the minimum inhibitory concentration (MIC) refers to the minimum drug concentration that can inhibit the growth of bacteria in a certain medium, which is the minimum inhibitory concentration. The minimum bactericidal concentration (MBC) is the minimum concentration required to kill 99.9% of bacteria. CSBTA had inhibitory and bactericidal effects against nine common Gram-positive and Gram-negative bacteria with MIC values ranging from 16.8–130 mg/mL [74]. Additionally, CSBTA exhibited antimicrobial activity with an MIC value of 20 mg/mL against *Staphylococcus aureus*. Notably, CSBTA combined with penicillin, cefradine, and levofloxacin effectively inhibited *S. aureus* [75]. The results above indicated that CSB and penicillin, cefradine, and levofloxacin had a synergistic antibacterial effect.

#### 6.9.2. Isolated Phytochemicals

The content of dehydrocavidine (**1**) from CSB is high, approximately 0.1–0.2%. Dehydrocavidine (**1**) has inhibitory effects on *S. aureus*, *Streptococcus β-haemolyticus*, diphtheria bacilli, and β-streptococcus, as well as penicillin-resistant *Staphylococcus albicans* and *S. aureus* [21,68]. In addition, in vitro antibacterial tests further showed that dehydrocavidine (**1**) had certain inhibitory effects on Gram-positive strains at a minimum concentration of 0.078 mg/mL. However, it had no inhibitory effects on Gram-negative bacteria [76]. Palmatine (**9**) has a strong inhibitory effect on some common pyogenic cocci and intestinal pathogenic bacteria [81]. The potential mechanism of the antibacterial activity of these compounds needs to be studied in the future.

### 6.10. Choleretic Effects

#### Crude Extract

The impairment of bile flow generally results from a decreased function of the liver and gallbladder. Many traditional Chinese medicines are considered to have good choleretic effects. A researcher found that intravenous injection of CSBTA could increase bile excretion in normal Sprague–Dawley (SD) rats [77], which suggested that CSBTA has a regulating protection for the function gallbladder. However, intravenous injection of CSBTA (20 mg/kg) had no cholagogic effect on anesthetized guinea pigs [66]. Its mechanism needs further elaboration.

### 6.11. Other Activities

Without affecting the normal physiological state of rats, CSBTA can significantly reduce the levels of plasma total cholesterol and low-density lipoprotein cholesterol in rats, regulate the level of blood lipids in rats fed a high-fat diet, and protect against fatty liver diseases caused by a high-fat diet [79]. Based on 16S rRNA gene sequencing and untargeted metabolomics analyses, CSBTA is an effective and reliable compound for use in co-metabolism and intestinal flora intervention methods in rats with antibiotic-induced intestinal flora imbalance [78]. Regarding the intestinal mucosal transport of dehydrocavidine (**1**), it had the ability to be effluxed by transporters, but it showed the characteristics of passive transport at a higher concentration range [80]. Apart from the mentioned activities, we suggest that we should further explore other pharmacological properties of CSB.

## 7. Clinical Applications

### 7.1. Icteric Hepatitis

Icteric hepatitis is an injury to the hepatocytes caused by various factors. The destruction of liver tissue leads to a decrease in bilirubin uptake and a decrease in the binding functions of hepatocytes, which leads to an increase in bilirubin in the blood [82,83,84]. The common causes of icteric hepatitis are viral infection, drug injury, alcohol injury, autoimmune factors, and so on. A large number of studies have reported that CSBI has a high total effective rate in the treatment of icteric hepatitis (Table 3).

After treating some patients with chronic liver disease and jaundice with CSBI, the symptoms of fatigue, abdominal distension, liver pain, and anorexia were significantly improved [84], and the decreases in transaminase and bilirubin were more obvious [82]. However, after CSBI injection, a patient with acute icteric hepatitis developed symptoms of pruritus, palpitation, chills, and fever, and immediately discontinued the use of CSBI. After discontinuation, the patient did not develop these symptoms again, which were considered to be an allergic reaction to CSBI [83]. All in all, CSBI plays a significant role in acute and chronic hepatitis with a higher total effective rate, but a few adverse reactions occur. Additionally, how to solve those adverse reactions is a potential research point.

In the clinical symptoms and the liver function indexes of some icteric hepatitis, CSBI injection is found to be superior to potassium magnesium aspartate [12,88,89,118], “Yinzhihuang” injection (including baicalin and the extracts of *Artemisiacapillaris* Thunb, *Gardenia jasminoides* Ellis, and *Lonicera japonica* Thunb.) [87]. Meanwhile, the curative effect of CSBI plus sugar once a day was significantly higher than that of 40 mL of Xilikang plus sugar once a day [85]. CSBI can repair liver damage after interventional therapy for advanced liver cancer with hepatocellular jaundice without obvious adverse reactions [86].

In some cases, CSB combined with other drugs for the treatment of icteric hepatitis showed high total effective rate, such as “Yinzhihuang” injection [119], compound glycyrrhizin [120], “Danshen” (*Salvia miltiorrhiza* Bge), and “Yinzhihuang” [90,91]. In addition, *Artemisiae Scopariae* Herba (30 g), CSB (30 g), *Lysimachia christinae* Hance (30 g), *Serissa japonica* (30 g), *Pteris multifida poir* (10 g), and *Ardisia japonica* (10 g) were used in the treatment of acute icteric hepatitis, The contents of TB and ALT decreased compared with those before treatment. Taking these results into account, the combined drug therapy with CSBI has a high efficacy in the treatment of icteric hepatitis in spite of the unknown metabolic interactions and mechanisms of action.

### 7.2. Viral Hepatitis

Viral hepatitis is a group of infectious diseases caused by hepatitis viruses that mainly damage the liver. The clinical manifestations of various hepatitis diseases are similar, as most patients may have digestive tract symptoms, systemic symptoms, jaundice, right upper abdominal pain, and so on.

In some clinical studies, after being treated with CSBI, ALT, and TBIL, levels were significantly decreased in patients with viral hepatitis [93] and chronic viral hepatitis B [92]. Compared with “Yinzhihuang” injection, CSBI has a better effect on patients with acute and chronic viral hepatitis and significantly decreased the levels of ALT, AST, and TBIL [94]; the degrees of reduction in TBIL, DBIL, and ALT were also significantly different [95]. Additionally, compared with diammonium glycyrrhizinate injection, CSBI has more obvious efficacy in the symptoms of jaundice. However, two patients reported that they had local vascular pain during the CSBI infusion, but the pain disappeared after the infusion was completed [96].

CSBI was shown to effectively improve the clinical symptoms of patients with acute and chronic viral hepatitis, and decrease the levels of serum total bilirubin (STB), 1-min bilirubin, ALT, and AST in a short period of time [97]. The effect of CSBI on STB was significantly better than that of potassium magnesium aspartate after 4 weeks of treatment [121]. In addition, “Danshen” injection combined with CSBI was used to treat severe jaundice stemming from viral hepatitis with a total effective rate of 96.14%. There was one mild rash in the treatment group and another in the control group [98]. From the perspective of enzymes, CSBI treats acute and chronic viral hepatitis by reducing the expression of ALT, AST, TBIL, and DBIL.

### 7.3. Acute and Chronic Hepatitis

Hepatitis is inflammation of the liver caused by various factors and is divided into acute and chronic categories. Acute hepatitis is an abnormal liver function that occurs for the first time.

In a study, the levels of serum TBIL and ALT in patients with chronic hepatitis B treated with CSBI were found to be significantly lower than those in the control group, and symptoms such as fatigue, nausea, vomiting, and greasiness were improved [92,122]. In addition, injection of CSBI at the “Zusanli” acupoint can not only treat patients with acute hepatitis but also play the role of acupuncture in dredging meridians, regulating visceral function, strengthening body resistance, and eliminating toxins. Therefore, the combination of the two treatments has a good therapeutic effect; this provides an innovative perspective for the application of CSBI.

Magnesium isoglycyrrhizinate combined with CSBI was used to treat patients with chronic cholestatic hepatitis B [99]. Moreover, CSBI was combined with telbivudine for the treatment of chronic severe hepatitis B [123]. CSBI combined with breviscapine injection for the treatment of chronic severe hepatitis showed that the clinical efficacy in the treatment group was significantly better than that in the control group [100]. Furthermore, it has also been reported that CSBI can be combined with “Danshen” injection or atomolan for the treatment of chronic hepatitis B [101,124]. The above CSBI in combination with other drugs has a significant effect on the treatment of hepatitis B. Hence, assessing the contribution of combined drugs would optimize the combination and facilitate the development of CSBI-related drugs.

### 7.4. Liver Cancer

Liver cancer refers to malignant tumours occurring in the liver, usually due to primary liver cancer, but including both primary and metastatic liver disease. Primary carcinoma in liver cancer refers to carcinomas occurring in liver cells or intrahepatic bile duct cells. The etiology of primary liver cancer is not completely clear and may be the result of synergistic effects from multiple factors.

The effect of CSBI combined with interventional radiotherapy on liver function and quality of life (QOL) score in the observation group of a study were significantly higher than those in the control group after treatment [102,103]. Additionally, CSBI combined with interventional radiotherapy can effectively improve the clinical efficacy of middle and advanced liver cancer [104,105,107,108], and improve the quality of life of patients with advanced liver cancer [109]. In addition, CSBI had an obvious liver reparative effect after interventional therapy for advanced liver cancer with hepatocellular jaundice with fewer adverse reactions [86,106]. Hence, the combination therapies with CSBI and interventional radiotherapy have been shown to effectively treat liver cancer which encourages medicinal chemistry to further develop and improve therapies.

CSBI combined with hepatic arterial chemoembolization (TACE) significantly improved the immune function and quality of life of patients with advanced liver cancer [110]. CSBI can effectively prevent and treat liver damage after TACE treatment [112]. Moreover, CSBI combined with interventional therapy has a significant effect on relieving clinical symptoms and improving liver function, especially with respect to increasing serum albumin levels in patients with liver cancer [111]. Octreotide combined with CSBI had a significant effect on advanced liver cancer [113], No adverse reactions were observed except for local pain during subcutaneous injection of octreotide, nausea, vomiting, and local vascular prickling when CSBI was added to the treatment [125].

### 7.5. Hyperbilirubinemia

Bilirubin is a bile pigment that is not only the main pigment in human bile but also an important index of liver function and an important basis for the diagnosis of jaundice [126]. Jaundice may cause the patient’s skin, mucous membranes, and sclera to display yellow staining that may also be accompanied by abdominal distension, abdominal pain, loss of appetite, and other symptoms, some of which are even life-threatening in severe cases [114].

It has been reported that 54 cases of hyperbilirubinemia treated by CSBI showed significant clinical effects, with a total effective rate of 91.47% [126]. CSBI treatment can significantly reduce the degree of neonatal jaundice and shorten its duration, and adverse reactions such as rash and diarrhea were significantly lower in the CSBI treatment group than in the control group [114]. Furthermore, CSBI combined with reduced glutathione is effective in the treatment of hyperbilirubinemia.

### 7.6. Others

In addition to treating hepatitis and liver cancer, CSBI can also treat a number of other diseases. For example, CSBI has a good clinical effect in the treatment of advanced rectal cancer [115]. When ribavirin and CSBI were combined for the treatment of hemorrhagic fever with renal syndrome, the recovery of AST, ALT, LDH, and BUN levels was more significant [116]. Both CSBI and “Shengmai” injections (the main ingredients include *Radix ginseng rubra*, *Ophiopogon japonicus* (Linn. f.) Ker-Gawl. and *Schisandra chinensis*) had good curative effects in alleviating the symptoms of liver cirrhosis, protecting the liver, and enhancing the gallbladder. However, “Shengmai” injection was notably superior to CSBI in alleviating discomfort in the liver area and yellow skin staining, and especially in reducing serum transaminase and bilirubin [117]. Therefore, in order to fully utilize CSBI, researchers should further explore CSBI alone or in combination with other drugs to treat other diseases.

## 8. Toxicity Assessment

To date, research on the toxicology and safety of CSB is very limited from both a traditional and modern standpoint.

There are some studies on the toxicity of CSBTA [54,68,83,91], and CSBTA (20 mg/kg/day) may have an inhibitory effect on the growth of rats, which was more obvious after three weeks of administration. In addition, there were no significant changes in SGPT, NPN, or haemogram in dogs after administration of CSBTA (3 mg/kg/day) for 2 or 4 weeks. The animals in both the treatment and control groups showed different degrees of sinus arrhythmias before and after administration [66]. The LD_50_ of CSBTA administered subcutaneously to mice was determined to be 233 mg/kg [65]. In another study, after oral administration of the CSB extract (560, 450, 360, 290, and 230 mg/kg), most of the mice died from poisoning within 2 days. The LD_50_ of this CSB extract was 298.5 mg/kg, and the 95% confidence limit was 257.2–346.5 mg/kg [54]. Therefore, this CSBI extract can be considered safe for use in a certain dose range.

There has been only one report on the toxicity of the individual CSB compounds. Acute toxicity from dehydrocavidine manifests as lethargy, weakness, paralysis, and death. The LD_50_ was determined to be 71.6 ± 2.92 mg/kg. A subtoxicity test administered this compound at dosages of 10 and 5 mg/kg/day. [68]. On the basis of the literature, toxicity tests of pure substances in CSB can be rarely found, which means it is urgent to perform a more toxicological experiment on it to subsequently guide its clinical application.

In several clinical studies using CSBI, adverse reactions have occurred. When CSBI was used in a clinical study for the treatment of jaundice hepatitis, one patient developed pruritus, palpitations, chills, and fever [83]. In another report, three patients had pain at the injection site after intravenous administration of CSBI, but the symptoms disappeared after deceleration [91]. Moreover, diarrhoea and rash occurred during the treatment of hyperbilirubinemia [114].

When CSBI was used to treat viral hepatitis, some studies showed adverse reactions. One patient developed a low fever within a few hours after the first treatment, but there was no fever after the second treatment when the infusion speed was slowed down [94]. Two patients reported local vascular pain during CSBI infusion, with the pain disappearing after the infusion [96]. Additionally, one patient in the CSBI treatment group had a mild rash [98]. The adverse reaction related to the allergic reaction is caused by the specific constitution of the patient, suggesting that the doctor would pay attention to it.

Finally, there are adverse symptoms associated with the use of CSBI for the treatment of liver cancer. In one study, fever, trembling, and itchy skin occurred in the CSBI injection group (6.7%) [86,106]. In this study group, adverse reactions occurred in five patients, with an incidence rate of 10.42% [113]. In the treatment of patients with advanced liver cancer, the adverse effect rates in the treatment groups were 20.8%, 5.2%, and 5.2% [108,113,125].

## 9. Conclusion and Future Perspectives

In summary, the botanical distribution and description, traditional usages, phytochemistry, pharmacological activities, and clinical applications of CSB are presented in the current review. Although the related quality control standards of CSB have not been included in the Chinese Pharmacopoeia, according to ancient Chinese herbal writings, CSB has been widely applied for treating hepatitis, liver cancer, clearing away heat, and toxic materials, relieving pain, and stopping bleeding. Alkaloids, the abundant active ingredients of CSB based on the current phytochemical, pharmacological studies, and clinical applications, have the fruitful effects of protecting the liver, being an anti-hepatitis virus, and having anti-tumour and analgesic effects, which expand traditional Chinese medicine applications.

At present, studies on the chemical constituents, pharmacological activities, and clinical applications of CSB have made substantial development whereas the scientific research gap in the mechanism of action remains. Therefore, to improve the safety and efficacy of CSB, further exploration and discussion of relative studies on CSB from the following aspects in the future are required.

Firstly, only 80 different secondary metabolites have been reported with 50% alkaloids, when the compressive structural activity relationship between the structural features and bioactivities is highly required. In addition, most major studies have been focused on the exploration of alkaloids from CSB, and studies on other bioactive components are limited which may lead to the loss of significant discoveries.

Secondly, diverse pharmacological effects of CSB have been reported whereas limited studies have been conducted on its pharmacological mechanism of action. Thus, the pharmacological mechanism should be further elucidated. Moreover, the preliminary pharmacokinetic parameters of CBSTA have been accomplished by performing analytical studies of in vivo experiments. The clarified process in vivo and the reasonably modified dosage form of CSB would improve its usage of it with more wide and safe use in clinical practice.

Thirdly, combination drug developments with CSBI have been adopted in the clinical treatment of acute and chronic diseases, resulting in a vastly enriched the diversity of drug ratio between CSBI and drugs. However, the mechanism of the pharmacological action of combination and the contribution of each active component are remaining which require more effort and endeavors from chemists and biologists.

Fourthly, the main chemical alkaloid constituent of CSB has been determined to be dehydrocavidine. CSBI and tablets can be applied to clinical practices; however, clinical application of dehydrocavidine has not yet been reported.

Finally, non-adequate toxicological properties of CSB suggest that research should be performed assessing adverse effects and exploring potential toxicity components of CSB, which encourages that the safety evaluation should be required in each pharmacological study as a supplement in the future.

## Figures and Tables

**Figure 1 ijms-24-01626-f001:**
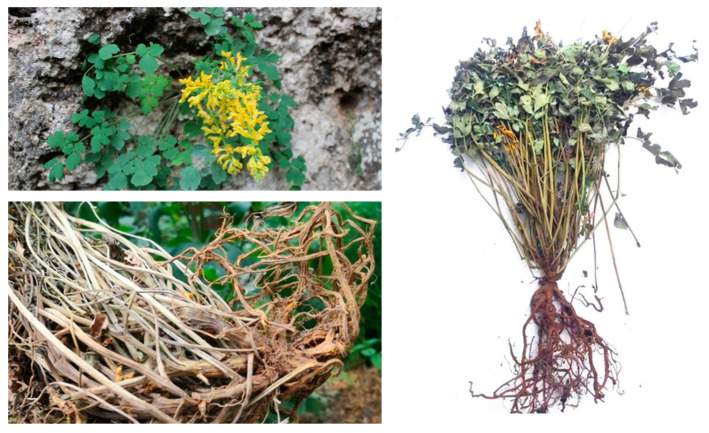
The pictures of *Corydalis saxicola* Bunting.

**Figure 2 ijms-24-01626-f002:**
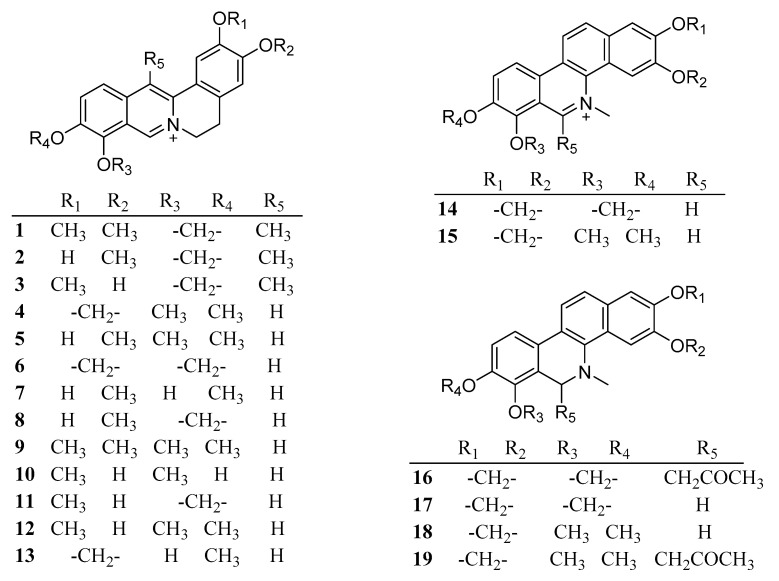
Chemical structures isolated from *Corydalis saxicola* Bunting.

**Table 1 ijms-24-01626-t001:** The phytoconstituents isolated from *Corydalis saxicola* Bunting.

NO.	Name	Molecular Formula	Molecular Weight	Plant Parts	References
1	dehydrocavidine	C_21_H_20_NO_4_	350.14	Whole plant	[17]
2	dehydroapocavidine	C_20_H_18_NO_4_	336.12	Root	[18]
3	dehydroisoapocavidine	C_20_H_18_NO_4_	336.12	Whole plant	[15]
4	berberine	C_20_H_18_NO_4_	336.12	Whole plant	[17]
5	dehydroisocorypalmine	C_20_H_20_NO_4_	338.14	Whole plant	[15]
6	coptisine	C_19_H_14_NO_4_	320.09	Root	[18]
7	tetradehydroscoulerine	C_19_H_18_NO_4_	324.12	Whole plant	[15]
8	dehydrocheilanthifoline	C_19_H_16_NO_4_	322.11	Whole plant	[19]
9	palmatine	C_21_H_22_NO_4_	352.15	Root	[18]
10	dehydrodiscretamine	C_19_H_18_NO_4_	324.12	Whole plant	[19]
11	thalifaurine	C_19_H_16_NO_4_	322.11	Root	[18]
12	jatrorrhizine	C_20_H_20_NO_4_	338.14	Whole plant	[13]
13	berberrubine	C_19_H_16_NO_4_	322.11	Whole plant	[20]
14	sanguinarine	C_20_H_14_NO_4_	332.09	Whole plant	[6]
15	chelerythrine	C_21_H_18_NO_4_	348.12	Whole plant	[17]
16	6-acetonyl-5,6-dihydrosanguinarine	C_23_H_19_NO_5_	389.13	Root	[18]
17	dihydrosanguinarine	C_20_H_15_NO_4_	333.10	Root	[18]
18	dihydrochelerythrine	C_21_H_19_NO_4_	349.13	Root	[18]
19	8-acetonyldihydrochelerythrine	C_24_H_23_NO_5_	405.16	Whole plant	[6]
20	1-nitro-apocavidine	C_20_H_20_N_2_O_6_	384.13	Whole plant	[14]
21	2, 9, 10-thrihydroxy-3-methoxytetrahydroprotoberberine	C_18_H_19_NO_4_	313.13	Whole plant	[16]
22	sinactine	C_20_H_21_NO_4_	339.15	Whole plant	[16]
23	(−)-13β-hydroxystylopine	C_19_H_17_NO_5_	339.34	Whole plant	[17]
24	tetrahydrocolumbamine	C_20_H_23_NO_4_	341.16	Whole plant	[17]
25	canadine	C_20_H_21_NO_4_	339.15	Whole plant	[2]
26	tetrahydropalmatrubine	C_20_H_23_NO_4_	341.16	Whole plant	[16]
27	(−)-2,9-dihydroxyl-3,11-dimethoxy-1,10-dinitrotetrahydroprotoberberine	C_19_H_19_N_3_O_8_	417.12	Whole plant	[6]
28	(+)-tetrahydropalmatine	C_21_H_25_NO_4_	355.15	Whole plant	[17]
29	(−)-corynoxidine	C_21_H_25_NO_5_	371.17	Whole plant	[6]
30	(−)-scoulerine	C_19_H_21_NO_4_	327.15	Whole plant	[17]
31	(−)-cavidine	C_21_H_23_NO_4_	353.16	Root	[18]
32	(+)-4-nitroisoapocavidine	C_20_H_20_N_2_O_6_	384.13	Whole plant	[6]
33	(+)-1-nitroapocavidine	C_20_H_20_N_2_O_6_	384.13	Whole plant	[6]
34	(±)-cavidine	C_21_H_23_NO_4_	353.16	Whole plant	[17]
35	(+)-thalictrifoline	C_21_H_23_NO_4_	353.16	Whole plant	[21]
36	(+)-cheilanthifoline	C_19_H_19_NO_4_	325.13	Whole plant	[16]
37	corydaline	C_22_H_27_NO_4_	369.19	Root	[18]
38	stylopine	C_19_H_17_NO_4_	323.12	Root	[18]
39	mesotetrahydrocorysamine	C_20_H_19_NO_4_	337.13	Whole plant	[16]
40	apocavidine	C_20_H_21_NO_4_	339.15	Whole plant	[16]
41	13-β-hydroxystylopine	C_19_H_17_NO_5_	339.11	Whole plant	[16]
42	(−)-tetrahydropalmatine	C_21_H_25_NO_4_	355.18	Whole plant	[19]
43	protopine	C_20_H_19_NO_5_	353.13	Whole plant	[17]
44	allocryptopine	C_21_H_23_NO_5_	369.16	Whole plant	[20]
45	berbinium	C_19_H_16_NO_4_^+^	322.11	Whole plant	[15]
46	1-formyl-5-methoxy-6-methylindoline	C_11_H_13_NO_2_	191.09	Whole plant	[15]
47	1-formyl-2-hydroxy-5-methoxy-6-methylindoline	C_11_H_13_NO_3_	207.09	Whole plant	[15]
48	saxicolaline A	C_20_H_22_NO_5_	356.15	Root	[18]
49	(+)-magnoflorine	C_20_H_24_NO_4_	342.17	Root	[18]
50	(+)-isocorydine	C_20_H_23_NO_4_	341.16	Whole plant	[19]
51	(−)-pallidine	C_19_H_21_NO_4_	327.15	Whole plant	[19]
52	*N*-methylnarceimicine	C_22_H_22_NO_8_	428.13	Root	[18]
53	adlumidine	C_20_H_17_NO_6_	367.11	Root	[18]
54	(−)-salutaridine	C_19_H_21_NO_4_	327.15	Root	[18]
55	cavidilinine	C_19_H_13_NO_4_	319.08	Whole plant	[20]
56	corypalline	C_11_H_15_NO_2_	193.11	Whole plant	[19]
57	oxyacanthine	C_37_H_40_N_2_O_6_	608.29	Whole plant	[22]
58	corydalisin A	C_37_H_38_N_2_O_9_	654.26	aerial parts	[1]
59	corydalisin B	C_38_H_40_N_2_O_10_	684.27	aerial parts	[1]
60	cannabisin F	C_36_H_36_N_2_O_8_	624.25	aerial parts	[1]
61	corydalisin C	C_38_H_40_N_2_O_10_	684.27	aerial parts	[1]
62	cannabisin D	C_36_H_36_N_2_O_8_	624.25	aerial parts	[1]
63	1,2-dihydro-6,8-dimethoxy-7-hydroxyl-1-(3,5-dimethoxy-4-hydroxyphenyl)-N1, N2-bis[2-(4-hydroxyphenyl)ethyl]-2,3-naphthalene dicarboxamide	C_38_H_40_N_2_O_10_	684.27	aerial parts	[1]
64	cannabisin E	C_36_H_38_N_2_O_9_	642.26	aerial parts	[1]
65	grossamide	C_36_H_36_N_2_O_8_	624.25	aerial parts	[1]
66	feruloylagmatine	C_15_H_22_N_4_O_3_	306.17	Whole plant	[19]
67	corysaxicolaine A	C_39_H_30_N_2_O_8_	654.20	aerial parts	[7]
68	β-sitosterol	C_29_H_50_O	414.39	Whole plant	[22]
69	daucosterol	C_35_H_60_O_6_	576.44	Whole plant	[22]
70	betulin	C_30_H_50_O_2_	484.43	Whole plant	[22]
71	betulinic acid	C_30_H_48_O_3_	498.41	Whole plant	[22]
72	β-amyrin	C_30_H_50_O	426.39	Whole plant	[20]
73	β-amyrin acetate	C_32_H_52_O_2_	468.40	Whole plant	[22]
74	(+)-oleanolic acid	C_30_H_48_O_3_	454.38	Whole plant	[22]
75	cholesterol	C_27_H_46_O	386.35	Whole plant	[23]
76	cycloeucalenol	C_30_H_50_O	426.39	Whole plant	[22]
77	5-hydroxy-3′, 4′, 6, 7-tetramethoxyflavone	C_19_H_18_O_7_	358.11	Whole plant	[20]
78	quercetin-3-O-β-D-galactoside	C_21_H_20_O_12_	461.10	Whole plant	[20]
79	tetra-amino-27 alkane	C_27_H_57_N	395.45	Whole plant	[23]
80	tetra-amino-28 alkane	C_28_H_59_N	409.46	Whole plant	[23]
81	uracil	C_4_H_4_N_2_O_2_	112.03	Whole plant	[20]

**Table 2 ijms-24-01626-t002:** Pharmacological activities of *Corydalis saxicola* Bunting.

	Tested Substance	Assay, Organism, or Cell Line	Biological Results	References
Anticancer activity	CSBTA	Walker 256 induced bone pain and osteoporosis in rats, Breast Cancer Cells, RAW 264.7 macrophage cells, Walker 256 cells	Improved bone pain and osteoporosis in rats, suppressed expression of Rankl, down regulated the ratio of RANKL/OPG, inhibited pathways of NF-κB and c-Fos/NFATc1 to suppressed osteoclast formation	[25]
	CSBTA	A549 cells	Inhibition migration of A549 cells by suppressing Cdc42 or Vav1	[8]
	CSBTA	A549 cells	Inhibited tumor growth by down-regulating Survivin, reduced the degree of bone destruction	[26]
	CSBTA	A549 cells	Inhibited the migration ability of A549 cells, decreased the expression of Cdc42 protein	[27]
	CSBTA	A549 cells	Inhibition proliferation, induced apoptosis and up regulation of caspase and down of survivin	[28]
	CSBTA and cis-platinum	A549 cells	Inhibition proliferation	[29]
	CSBTA	Tca8113 cells	Induced apoptosis and suppression of Bcl-2	[30]
	CSBTA	Tca8113 cells	Inhibition proliferation, induced apoptosis	[31]
	CSBTA	Tca8113 cells	Inhibition proliferation and telomerase activity	[32]
	CSBTA	CNE-1	112.41 µg/mL (IC_50_)	[23]
	CSBTA	CNE-2	123.46 µg/mL (IC_50_)	[23]
	CSBTA	A2780	148.40 µg/mL (IC_50_)	[23]
	CSBTA	SKOV3	128.51 µg/mL (IC_50_)	[23]
	CSBTA	PM2	166.66 µg/mL (IC_50_)	[23]
	CSB injection	mouse sarcoma S180	The average tumor inhibition rate was more than 30%	[33]
	CSB injection	ehrlich ascites tumor	The average tumor inhibition rate was more than 30%	[33]
	CSB injection and aristolochic acid	mouse sarcoma S180	The average tumor inhibition rate was more than 50%	[33]
	CSB injection and aristolochic acid	ehrlich ascites tumor	The average tumor inhibition rate was more than 50%	[33]
	CSB injection	mouse sarcoma S180	Some inhibition and inhibited respiration	[17]
	CSB injection	ehrlich ascites tumor	Some inhibition and inhibited respiration	[17]
	CSB injection	liver cancer	Some inhibition and inhibited respiration	[17]
	CSB injection	ascites cancer	Some inhibition and inhibited respiration	[17]
	CSB injection	rat sarcoma 256	Some inhibition and inhibited respiration	[17]
	CSB injection	mouse peritoneal macrophages	Significantly enhanced phagocytosis	[17]
	CSB injection	S180	Inhibition of respiratory metabolism	[34]
	CSB injection	HAC	Inhibition of respiratory metabolism	[34]
	CSB injection	EAC	Inhibition of respiratory metabolism	[34]
	CSB injection	S180, HAC, EAC, W256, rat	Significantly inhibition	[35]
	CSB injection	S180, HAC, EAC, W256	Killing effect	[36]
	aqueous extract	HepG2	Inhibition proliferation and migration	[25]
	Corydalisin A	MGC-803	83.56 ±1.89 μM (IC_50_)	[1]
	Corydalisin A	HepG2	> 100 μM (IC_50_)	[1]
	Corydalisin A	T24	> 100 μM (IC_50_)	[1]
	Corydalisin A	NCI-H460	> 100 μM (IC_50_)	[1]
	Corydalisin A	Spca-2	> 100 μM (IC_50_)	[1]
	Corydalisin B	MGC-803	> 100 μM (IC_50_)	[1]
	Corydalisin B	HepG2	> 100 μM (IC_50_)	[1]
	Corydalisin B	T24	> 100 μM (IC_50_)	[1]
	Corydalisin B	NCI-H460	> 100 μM (IC_50_)	[1]
	Corydalisin B	Spca-2	> 100 μM (IC_50_)	[1]
	Corydalisin C	MGC-803	8.81 ±2.05 μM (IC_50_)	[1]
	Corydalisin C	HepG2	22.23 ±1.85 μM (IC_50_)	[1]
	Corydalisin C	T24	9.62 ±1.46 μM (IC_50_)	[1]
	Corydalisin C	NCI-H460	25.79 ±1.04 μM (IC_50_)	[1]
	Corydalisin C	Spca-2	17.28 ±1.29 μM (IC_50_)	[1]
	Cannabisin F	MGC-803	10.10 ±1.15 μM (IC_50_)	[1]
	Cannabisin F	HepG2	38.93 ±1.07 μM (IC_50_)	[1]
	Cannabisin F	T24	11.54 ±1.49 μM (IC_50_)	[1]
	Cannabisin F	NCI-H460	30.96 ±1.27 μM (IC_50_)	[1]
	Cannabisin F	Spca-2	22.23 ±1.44 μM (IC_50_)	[1]
	Cannabisin E	MGC-803	> 100 μM (IC_50_)	[1]
	Cannabisin E	HepG2	> 100 μM (IC_50_)	[1]
	Cannabisin E	T24	46.54 ±1.62 μM (IC_50_)	[1]
	Cannabisin E	NCI-H460	> 100 μM (IC_50_)	[1]
	Cannabisin E	Spca-2	> 100 μM (IC_50_)	[1]
	Cannabisin D	MGC-803	> 100 μM (IC_50_)	[1]
	Cannabisin D	HepG2	> 100 μM (IC_50_)	[1]
	Cannabisin D	T24	> 100 μM (IC_50_)	[1]
	Cannabisin D	NCI-H460	> 100 μM (IC_50_)	[1]
	Cannabisin D	Spca-2	> 100 μM (IC_50_)	[1]
	1,2-dihydro-6,8-dimethoxy-7-hydroxy-1-(3,5-dimethoxy-4- hydroxyphenyl)-N 1, N 2 -bis [2-(4-hydroxyphenyl) ethyl]-2,3-naphthalene dicarboxamide	MGC-803; HepG2;T24; NCI-H460; Spca-2	55.16 ±0.78 μM (IC_50_); > 100 μM (IC_50_); 48.15 ±1.09 μM (IC_50_); > 100 μM (IC_50_); 43.89 ±1.57 μM (IC_50_)	[1]
	grossamide	MGC-803	26.95 ±1.24 μM (IC_50_)	[1]
	grossamide	HepG2	40.75 ±0.88 μM (IC_50_)	[1]
	grossamide	T24	21.19 ±1.53 μM (IC_50_)	[1]
	grossamide	NCI-H460	36.38 ±1.39 μM (IC_50_)	[1]
	grossamide	Spca-2	27.22 ±1.72 μM (IC_50_)	[1]
	Dehydrocavidine	SMMC-7721	Significantly inhibition	[37]
	palmatine	SMMC-7721	Significantly inhibition	[37]
	Dehydrocavidine	Tca8113	Significantly inhibition, suppression of NF-kappa B, P50 and P60	[38]
	CSBTA	Tca8113	Significantly inhibition, suppression of NF-kappa B, P50 and P60	[38]
	Dehydrocavidine	Tca8113	Inhibition proliferation, telomerase activity and the expression of hTERT	[39]
	Pallidine	DNA topoisomerase I	Strong inhibitory effect on human DNA topoisomerase I	[19]
	scoulerine	DNA topoisomerase I	Strong inhibitory effect on human DNA topoisomerase I	[19]
	chelerythrine	unknown	Have certain anticancer effect	[21]
	(−)-13β-hydroxystylopine	unknown	Have certain anticancer effect	[21]
	Corysaxicolaine A	T24	7.63 μM (IC_50_)	[7]
	Corysaxicolaine A	A549	13.32 μM (IC_50_)	[7]
	Corysaxicolaine A	HepG2	12.39 μM (IC_50_)	[7]
	Corysaxicolaine A	MGC-803	9.98 μM (IC_50_)	[7]
	Corysaxicolaine A	SKOV3	12.36 μM (IC_50_)	[7]
Hepatoprotective effects	CSBTA	rats	Interventional treatment of chronic liver injury	[40]
	CSBTA	HSC-T6	Induced apoptosis and autophagy	[41]
	CSBTA	CYP450s in rats	CYP1A2 (IC_50_, 38.08 μg/mL; K_i_, 14.3 μg/mL), CYP2D1 (IC_50_, 20.89 μg/mL; K_i_, 9.34 μg/mL), CYP2C6/11 (IC_50_ for diclofenac and S-mephenytoin, 56.98 and 31.59 μg/mL; K_i_, 39.0 and 23.8 μg/mL), CYP2B1 (IC_50_, 48.49 μg/mL; )K_i_, 36.3 μg/mL)	[42]
	CSBTA	chronic hepatotoxicity in rats	Restored the levels of 2-oxoglutarate, citrate, hippurateand taurine	[43]
	CSBTA	acute hepatic injury rats	Significantly reduced the content of AST, ALT	[44]
		chronic hepatic injury rats	Significantly increased the level of serum TP, reduced the content of AST, ALT, AKP, LN and HA	[45]
	CSBTA	immune hepatic injury rat	Reduced serum GOT activity, IL-4, increased the rate of IFN-γ/IL-4	[46]
	CSBTA	rats	Have certain preventive and therapeutic effect on acute liver injury and on chronic liver fibrosis	[47]
	CSBTA	rats	Obvious protective effect on acute liver injury, inhibited the formation of chronic liver fibrosis	[48]
	CSBTA	hepatic fibrosis rats	Inhibited the expression of TGF-β1 and MMP-9	[49]
	CSBTA	acute hepatic injury rats	Increased the content of AST, ALT and SOD, reduced MDA	[50]
	aqueous extract	liver fibrosis in rats	Regulated the level of some amino acids, identified 157 potential targets of CS and265 targets of liver fibrosis	[51]
	aqueous extract	acute hepatic injury rats	Improved deviations of metabolites (soleucine, alanine, glutamine, phosphocholine and glycerol)	[52]
	aqueous extract	acute hepatic injury rats	Increased the content of AST, ALT and SOD, reduced MDA	[53]
	aqueous extract	acute hepatic injury rats	Reduced the contents of AST and ALTpromote the production of mouse hemolysin antibody LD_50_ = 298.5 mg·kg^-1^	[54]
	aqueous extract	acute hepatic injury rats	Reduced the content of AST and ALT	[55]
	Dehydrocavidine	HSC-T6	Inhibition proliferation, induced apoptosis	[56]
	palmatine	HSC-T6	Inhibition proliferation, induced apoptosis	[56]
	berberine	HSC-T6	Inhibition proliferation, induced apoptosis	[56]
	Dehydrocavidine	hepatic fibrosis rats	Reduced hepatic hydroxyproline, increases urinary hydroxyproline	[57]
	Dehydrocavidine	liver injury in rats	Down regulated EPHX2, HYOU1, GSTM3, Sult1a2 and P450, reduce free radical, lose weight, MDA, ALT, AST, ALP and TBIL	[58]
	Dehydrocavidine	liver injury in rats	Increased ALT, AST and TBIL, Reduces the inflammatory cell infiltration of cell degeneration and necrosis and damages the ultrastructure of liver cells	[59]
Anti-HBV activity	extract	Duck hepatitis B virus	Reduced DHBV-DNA	[60]
	total extract of root	HBsAg	0.17 mg/mL (IC_50_)	[18]
	total extract of root	HBeAg	<0.04 mg/mL (IC_50_)	[18]
	Saxicolalines A	HBsAg	2.19 μM (IC_50_)	[18]
	Saxicolalines A	HBeAg	>2.81μM (IC_50_)	[18]
	*N*-methylnarceimicine	HBsAg	1.22 μM (IC_50_)	[18]
	*N*-methylnarceimicine	HBeAg	1.84 μM (IC_50_)	[18]
	6-acetonyl-5,6-dihydrosanguinarine	HBsAg	6.55 μM (IC_50_)	[18]
	6-acetonyl-5,6-dihydrosanguinarine	HBeAg	>2.54 μM (IC_50_)	[18]
	dihydrochelerythrine	HBsAg	<0.05 μM (IC_50_)	[18]
	dihydrochelerythrine	HBeAg	<0.05 μM (IC_50_)	[18]
	adlumidine	HBsAg	1.35 μM (IC_50_)	[18]
	adlumidine	HBeAg	>2.73 μM (IC_50_)	[18]
	(−)-salutaridine	HBsAg	0.26 μM (IC_50_)	[18]
	(−)-salutaridine	HBeAg	0.43 μM (IC_50_)	[18]
	palmatine	HBsAg	>4.26 μM (IC_50_)	[18]
	palmatine	HBeAg	>4.26 μM (IC_50_)	[18]
	protopine	HBsAg	2.61 μM (IC_50_)	[18]
	protopine	HBeAg	>4.25 μM (IC_50_)	[18]
	coptisine	HBsAg	2.74 μM (IC_50_)	[18]
	coptisine	HBeAg	3.19 μM (IC_50_)	[18]
	(+)-magnoflorine	HBsAg	>4.39 μM (IC_50_)	[18]
	(+)-magnoflorine	HBeAg	>4.39 μM (IC_50_)	[18]
	dehydrocheilanthifoline	HepG2.2.15	115.95 μM (CC_50_)	[3]
	dehydrocheilanthifoline	HBsAg	15.84 ± 0.36 μM (IC_50_)	[3]
	dehydrocheilanthifoline	HBeAg	17.12 ± 0.45 μM (IC_50_)	[3]
	dehydrocheilanthifoline	Extracellular DNA	15.08 ± 0.66 μM (IC_50_)	[3]
	dehydrocheilanthifoline	Intracellular DNA	7.62 ± 0.24 μM (IC_50_)	[3]
	dehydrocheilanthifoline	Intracellular cccDNA	8.25 ± 0.43 μM (IC_50_)	[3]
	Crude extract	HBsAg	0.16 mg/mL (IC_50_)	[61]
	Crude extract	HBeAg	< 0.04 mg/mL (IC_50_)	[61]
	6-acetonyl-5, 6-dihydrosanguinarine	HBsAg	0.65 mg/mL (IC_50_)	[61]
	6-acetonyl-5, 6-dihydrosanguinarine	HBeAg	>1.00 mg/mL (IC_50_)	[61]
	dihydrochelerythrine	HBsAg	<0.02 mg/mL (IC_50_)	[61]
	dihydrochelerythrine	HBeAg	<0.02 mg/mL (IC_50_)	[61]
	adlumidine	HBsAg	0.50 mg/mL (IC_50_)	[61]
	adlumidine	HBeAg	>1.00 mg/mL (IC_50_)	[61]
	(−)-salutaridine	HBsAg	0.09 mg/mL (IC_50_)	[61]
	(−)-salutaridine	HBeAg	0.15 mg/mL (IC_50_)	[61]
	palmatine	HBsAg	>1.50 mg/mL (IC_50_)	[61]
	palmatine	HBeAg	>1.50 mg/mL (IC_50_)	[61]
	protopine	HBsAg	0.92 mg/mL (IC_50_)	[61]
	protopine	HBeAg	>1.50 mg/mL (IC_50_)	[61]
	coptisine	HBsAg	0.88 mg/mL (IC_50_)	[61]
	coptisine	HBeAg	>1.02 mg/mL (IC_50_)	[61]
	(+)-magnoflorine	HBsAg	>1.50 mg/mL (IC_50_)	[61]
	(+)-magnoflorine	HBeAg	1.50 mg/mL (IC_50_)	[61]
	dehydrocavidine	HBsAg	33% inhibition [62.5 μg/mL]	[15]
	dehydrocavidine	HBeAg	22% inhibition [62.5 μg/mL]	[15]
	dehydroapocavidine	HBsAg	39% inhibition [62.5 μg/mL]	[15]
	dehydroapocavidine	HBeAg	24% inhibition [62.5 μg/mL]	[15]
	dehydroisoapocavidine	HBsAg	29% inhibition [62.5 μg/mL]	[15]
	dehydroisoapocavidine	HBeAg	23% inhibition [62.5 μg/mL]	[15]
	berberine	HBsAg	8% inhibition [62.5 μg/mL]	[15]
	berberine	HBeAg	7% inhibition [62.5 μg/mL]	[15]
	dehydroisocorypalmine	HBsAg	6% inhibition [62.5 μg/mL]	[15]
	dehydroisocorypalmine	HBeAg	6% inhibition [62.5 μg/mL]	[15]
	coptisine	HBsAg	6% inhibition [62.5 μg/mL]	[15]
	coptisine	HBeAg	9% inhibition [62.5 μg/mL]	[15]
	tetradehydroscoulerine	HBsAg	7% inhibition [62.5 μg/mL]	[15]
	tetradehydroscoulerine	HBeAg	6% inhibition [62.5 μg/mL]	[15]
	berbinium	HBsAg	9% inhibition [62.5 μg/mL]	[15]
	berbinium	HBeAg	6% inhibition [62.5 μg/mL]	[15]
	1-formyl-5-methoxy-6-methylindoline	HBsAg	2% inhibition [62.5 μg/mL]	[15]
	1-formyl-5-methoxy-6-methylindoline	HBeAg	7% inhibition [62.5 μg/mL]	[15]
	1-formyl-2-hydroxy-5-methoxy-6-methylindoline	HBsAg	5% inhibition [62.5 μg/mL]	[15]
	1-formyl-2-hydroxy-5-methoxy-6-methylindoline	HBeAg	3% inhibition [62.5 μg/mL]	[15]
Enhancement of immune function	CSBTA	rats	CSBTA (40 μg/mL) began to enhance, enhanced the levels of T cell production of IFN-γ and IL-2	[62]
Antioxidant activity	CSBTA	rats	Reduced the of content MDA and increase SOD activity, enhance the antioxidant capacity of rat liver	[63]
	cavidine	DPPH assay	6.85 mg/mL (IC_50_)	[2]
	cheilanthifoline	DPPH assay	0.25 mg/mL (IC_50_)	[2]
	tetrahydropalmatine	DPPH assay	3.79 mg/mL (IC_50_)	[2]
	stylopine	DPPH assay	2.56 mg/mL (IC_50_)	[2]
	canadine	DPPH assay	2.18 mg/mL (IC_50_)	[2]
	dehydrocavidine	DPPH assay	16.51 mg/mL (IC_50_)	[2]
	dehydrocheilanthifoline	DPPH assay	1.63 mg/mL (IC_50_)	[2]
	berberine	DPPH assay	7.40 mg/mL (IC_50_)	[2]
	pallidine	DPPH assay	1.00 mg/mL (IC_50_)	[2]
Effects on the central nervous system	CSBTA	rats	Reduced the content of DOPAC, HVA, 5-HT and 5-HIAA, the level of DA has no effect (50, 100 mg/kg CSBTA)	[64]
	CSBTA	rats	Reduced activity (25 mg/kg CSBTA)	[65]
	CSBTA	monkey	Reduced activity (12 mg/kg CSBTA)	[65]
	CSBTA	cats	Reduced activity (10-15 mg/kg CSBTA)	[65]
	CSBTA	rats	Reduced irritated response (50 mg/kg CSBTA)	[65]
	CSBTA	rats	77% suppressed conditional emission (50 mg/kg CSBTA)	[65]
	CSBTA	rats	Increased the hypnotic time of pentobarbital sodium by more than 2 to 4 times (25 mg/kg CSBTA)	[65]
	CSBTA	rabbit	Activity slow down (20-30 mg/kg CSBTA)	[65]
	CSBTA	rats	LD_50_ = 223 mg/kg	[65]
	CSBTA	rats	Reduced the arthritis (50 mg/kg CSBTA)	[66]
	Dehydrocavidine	rats	Reduced the content of DA and HVA	[67]
	Dohydrocyaidine	rats	Reduced spontaneous activity	[68]
	Dohydrocyaidine	rats	Synergistic effect with barbiturates	[68]
Anti-inflammatory activity	CSBTA	M1 macrophages	Obvious toxic effect on the activity of M1-Mφ, significantly reduced the mRNA level of IL-6, TNF-α, CD86, IL-1β	[69]
	CSBTA	rats	Significantly inhibited the addition of capillary permeability, and suppressed exudation, edema and connective tissue hyperplasia	[70]
	*Corydalis saxicola* suppository	chronic pelvic inflammatory disease model rats	Significantly inhibited the uterine swelling, significantly reduced the spleen index, hemameba, neutrophil, TNF-α, IL-6 and MDA, improved thymus index, ovary index, lgG, lgM and SOD	[71]
	*Corydalis saxicola* rectal suppository	rats	Obvious inhibited ear swelling, the addition of capillary permeability, and writhing reaction in rat	[72]
Analgesic effect	CSBTA	rats	Reduced the level of proinflammatory cytokines, such as TNF-α, IL-1β and PGE2. inhibited the overexpression level of DRG, TG, p-p38 and TRPV1	[73]
	CSBTA	rats	Inhibited the "writhing reaction" in rat (50 mg/kg CSBTA), improve the "pain closure" of rats to heat stimulation (100 mg/kg CSBTA)	[66]
	Dohydrocyaidine	rats	The effects of sedative, analgesic, and spasmolysis, LD_50_ = 71.6±2.92 mg/kg	[68]
	deheydrocavidine	unknown	Have certain sedative and analgesic effects	[21]
Antibacterial activity	CSBTA	staphylococcus aureus	17.8 mg/mL (MIC)	[74]
	CSBTA	staphylococcus aureus	70.0 mg/mL (MBC)	[74]
	CSBTA	streptococcus pyogenes	20.5 mg/mL (MIC)	[74]
	CSBTA	streptococcus pyogenes	70.0 mg/mL (MBC)	[74]
	CSBTA	streptococcus faecalis	17.8 mg/mL (MIC)	[74]
	CSBTA		70.0 mg/mL (MBC)	[74]
	CSBTA	escherichia coli	35.5 mg/mL (MIC)	[74]
	CSBTA		70.0 mg/mL (MBC)	[74]
	CSBTA	pseudomonas aeruginosa	70.0 mg/mL (MIC)	[74]
	CSBTA		>300 mg/mL (MBC)	[74]
	CSBTA	shigella flexneri	16.8 mg/mL (MIC)	[74]
	CSBTA		35.5 mg/mL (MBC)	[74]
	CSBTA	salmonella typhi	35.5 mg/mL (MIC)	[74]
	CSBTA		70.0 mg/mL (MBC)	[74]
	CSBTA	salmonella enteritidis	16.8 mg/mL (MIC)	[74]
	CSBTA		70.0 mg/mL (MBC)	[74]
	CSBTA	klebsiella pneumoniae	70.0 mg/mL (MIC)	[74]
	CSBTA		130.0 mg/mL (MBC)	[74]
	CSBTA	proteus	70.0 mg/mL (MIC)	[74]
	CSBTA		130.0 mg/mL (MBC)	[74]
	CSBTA	candida albicans	130.0 mg/mL (MIC)	[74]
	CSBTA		> 300 mg/mL (MBC)	[74]
	extract	staphylococcus aureus	20.0 mg/mL (MIC)	[75]
	extract +Cefradine	staphylococcus aureus	0.375 (FIC) synergistic effect	[75]
	extract + Levofloxacin	staphylococcus aureus	0.5 (FIC) synergistic effect	[75]
	extract + Fosfomycin	staphylococcus aureus	1.5 (FIC) irrelevant effect	[75]
	extract + Penicillin	staphylococcus aureus	0.375 (FIC) synergistic effect	[75]
	dehydrocarvidine	gram-positive strains; gram-negative bacterium	Have certain inhibitory effect on gram-positive strains, minimum concentration is 0.078 mg/mL, and has no inhibitory effect on gram-negative bacteria	[76]
	deheydrocavidine	staphylococcus aureus; beta hemolytic streptococcus; corynebacterium diphtheriae; penicillin-resistant white staphylococcus aureus	Have an inhibiting effect	[21]
	dohydrocyaidine	rats	Antibacterial effect	[68]
ChOleretic effects	CSBTA	guinea pig	Bile secretion is temporarily reduced	[66]
	extract	rats	Increased the amount of bile excretion	[77]
Other activities	CSBTA	rats	Intervention of host co-metabolism and intestinal flora in rats with intestinal flora imbalance	[78]
	CSBTA	rats	Significantly decreased the levels of plasma total cholesterol and low-density lipoprotein cholesterol in rats, regulated blood lipid levels in high-fat diet rats	[79]
	Dehydrocarvidine	rats	The transport of dehydrocavidine was carried out in vitro at different intestine segments	[80]
	Dehydrocarvidine	Caco-2 cells	The bi-directional transport of dehydrocavidine in Caco-2 monolayer model showed significant difference	[80]

**Table 3 ijms-24-01626-t003:** The clinical application of *Corydalis saxicola* Bunting.

Class	Drugs	Cases	Result	Adverse Reaction	References
Icteric hepatitis	CSBI	unknow	The total effective rate of the treatment group was 87.5%, the control group was 62.5% (*p* < 0.01)		[85]
	CSBI	42	Significantly decreased contents of ALT, AST, γ-GT and TBIL (*p* < 0.05).		[86]
	CSBI	60	The clinical symptoms and liver function in the treatment group were better than those in the control group (*p* < 0.05)		[87]
	CSBI	82	Significantly improved symptoms of fatigue, abdominal distension, hepatalgia and poor appetite; obvious decrease of transaminase and bilirubin		[82]
	CSBI	98	Improvement rate of poor appetite, hepatalgia, fatigue and abdominal distension was 85.7%, 84.4%, 76.8% and 87.8%, respectively, summary improvement is 83.4%. Significantly decreased contents of ALT, TBIL		[88]
	CSBI	29	The total effective rate of the treatment group was 93.1%, the control group was 71.0% (*p* < 0.05)		[89]
	CSBI	1	Caused allergic reactions	Itching, palpitating, chills, fever	[83]
	Salvia miltiorrhiza Bge & Yinzhihuang & CSB	90	The total effective rate of the treatment group was 96.0%, the control group was 82.5% (*p* < 0.05)		[90]
	Salvia miltiorrhiza Bge & Yinzhihuang & CSB	90	The total effective rate of the treatment group was 96.0%, the control group was 82.5% (*p* < 0.05)	Precardiac discomfort, urticaria, skin itching, pain at the injection site	[91]
Viral hepatitis	CSBI	93	The total effective rate of the treatment group was 91.7%, the control group was 68.9% (*p* < 0.03)	There were no adverse reactions	[92]
	CSBI	50	Significantly decreased ALT and TBIL (*p* < 0.05), the total effective rate of the treatment group was 94.0%		[93]
	CSBI	208	TBil of treatment group dropped by 71.22%, the control group was 44.30% (*p* < 0.01); Dbil of treatment group dropped by 67.53% (*p* < 0.01)	low-grade fever	[94]
	CSBI	360	Significant difference in improvement rate of hepatalgia and poor appetite (*p* < 0.01), decreased the levels of T-BILI, D-BILI and ALT		[95]
	CSBI	60	The total effective rate of the treatment group was 88.23%, the control group was 76.92% (*p* < 0.05)	Local vascular pain	[96]
	CSBI	33	Decreased the contents of ALT and AST in a short time, improved protein metabolism		[97]
	Compound Danshen & CSB	100	The total effective rate of the treatment group was 96.14%, the control group was 64.58% (*p* < 0.01)	Mild rash	[98]
Acute and chronic hepatitis	CSBI	93	The total effective rate of the treatment group was 91.7%, the control group was 68.9% (*p* < 0.03)	There were no adverse reactions	[92]
	Magnesium isoglycyrrhizinate & CSB	65	The total effective rate of the treatment group was 89.23%, the control group was 70.14% (*p* < 0.05)		[99]
	CSB & Telbivudine	80	The total effective rate of the treatment group was 72.5%, the control group was 50% (*p* < 0.05)		[100]
	CSB & Danshen injection	70	Significantly decreased ALT and TBIL, increased the ratio of A/G		[101]
Liver cancer	CSBI	96	The total effective rate of the treatment group was 83.3%, the control group was 72.9% (*p* < 0.05), Significantly decreased ALT and AST		[102]
	CSBI	96	The total effective rate of the treatment group was 83.3%, the control group was 72.9% (*p* < 0.05), the QOL of the treatment group (68.6±7.2) more than the control group (60.5±6.1) after treatment (*p* < 0.05)		[103]
	CSBI	96	The total effective rate of the treatment group was 79.2%, the control group was 50.0% (*p* < 0.05), effectively improve the levels of serum IL-2, IFN-γ and TNF-α, better than the control group		[104]
	CSBI	96	The total effective rate of the treatment group was 81.3%, the control group was 70.8% (*p* < 0.05), after treatment, the INF-γ, IL-4 and the ratio of INF-γ/IL-4 in the observation group were significantly better than those in the control group (*p* < 0.05)		[105]
	CSBI	120	After treatment, the liver function recovery effect of the injection group was better than that of the control group (*p* < 0.05)	fever, skin itch; the injection group (6.7%), the control group (8.3%)	[106]
	CSBI	96	The total effective rate of the treatment group was 72.9%, the control group was 52.1% (*p* < 0.05)		[107]
	CSBI	96	The total effective rate of the treatment group was 83.3%, the control group was 64.4% (*p* < 0.05)	The adverse effects rate of the treatment group was 20.8%, the control group was 39.6% (*p* < 0.05)	[108]
	CSBI	96	The total effective rate of the treatment group was 97.92%, the control group was 87.50% (*p* < 0.05)	The adverse effects rate of the treatment group was 10.42%, the control group was 39.6% (*p* < 0.05)	[109]
	CSBI	110	Quality of life improvement rate in the treatment group was 81.82%, the control group was 52.73% (*p* < 0.05)		[110]
	CSBI	42	Significantly decreased the levels of ALT, AST, TBIL and γ-GT	Fever, shiver, skin itch	[86]
	CSBI	60	Significantly decreased the level of AFP, the white blood cell count goes up		[111]
	CSBI	46	Significantly increased the level of ALT and AST (*p* < 0.05), decreased TBIL (*p* > 0.05)		[112]
	CSBI & Octreotide	116	The total effective rate of the treatment group was 96.4%, the control group was 92.9% (*p* < 0.05)	The adverse effects rate of the treatment group was 5.2%, the control group was 17.2% (*p* < 0.05)	[113]
Hyperbilirubinemia	CSBI	126	The total effective rate of the treatment group was 95.3%, the control group was 93.6% (*p* > 0.05)	Diarrhea, rash	[114]
Others					
Rectal cancer	CSBI	68	The total effective rate of the treatment group was 38.24%, the control group was 5.88% (*p* < 0.05)		[115]
Hemorrhagic fever with renal syndrome	CSBI & Ribavirin	60	Significantly decreased the duration of fever period and oliguria period, obvious the recovery of AST, ALT, LDH and BUN than the control group (*p* < 0.05)		[116]
Liver cirrhosis	CSBI	60	Significantly effect in relieving symptoms, protecting liver and gallbladder		[117]

## Data Availability

Not Applicable.

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
