# Peer review of "Corydalis saxicola* Bunting: A Review of Its Traditional Uses, Phytochemistry, Pharmacology, and Clinical Applications"

_ijms, 2023, doi:10.3390/ijms24021626_

Round 1
Reviewer 1 Report
Dear authors,
In the following, I have mentioned several points that must be corrected.
In Abstract section,
- Please check your manuscript in terms of grammatical errors and typos.
-Please change the first two sentences of this section, and provide a more detailed and comprehensive introduction for this section.
- In Keywords section,
- Please mention all keywords in alphabetical order.
-Please try to provide each keyword up to two words.
In introduction section,
- If you can, please use the recently published papers as references in the introduction section (since 2018).
- Please try to extend this section and provide more information regarding the primary application of Corydalis saxicola Bunting (CSB).
- In my opinion, the introduction section has not a sufficient cohesion and coherence. Please add a more cohesive introduction regarding the primary available data about Corydalis saxicola Bunting (CSB).
-Please add a more detailed suggestion regarding the application of Corydalis saxicola Bunting (CSB) in the future.
-Please add a separate section as methods and explain your search strategy and search engines and also inclusion and exclusion criteria used in this review.
- Please write the following sentence as a separate paragraph (also, please write the aim of your study in more detail),
"This review lays a foundation for further study on the pharmacological actions, mechanisms and clinical applications of CSB."
In Botanical distribution and description section,
- In the second paragraph, please add a figure to show the appearance characteristics of Corydalis saxicola Bunting (CSB) as explained in this section.
In Pharmacology section,
- In anticancer activity section, please add a Table regarding a summary of different cancer cells that Corydalis saxicola Bunting (CSB) shows anticancer activity against them and add a separate reference for each one.
-Please try to provide a conclusion and suggestion for each section.
In Antioxidant activity section,
-Please add a description about the IC50 meaning and DPPH method and then introduce different results of previous studies.
-In Isolated phytochemicals section, please add references for the first two sentences.
In Effects on the central nervous system section,
-Please add an introduction about the effects of different plant extracts on nerves system related diseases and then explain the obtained results about Corydalis saxicola Bunting (CSB).
In Anti-inflammatory activity section,
-Please add a summary about inflammatory system and mechanisms that regulate inflammation and then explain the obtained results about Corydalis saxicola Bunting (CSB).
In Analgesic effect section,
-Please add an introduction about the Analgesic effect and different plants that have Analgesic effects and then explain the obtained results about Corydalis saxicola Bunting (CSB).
In Antibacterial activity section,
-Please add an introduction for this section and explain MIC and MBC meanings, then provide the results of previous studies. If possible, summarize the previous results as a Table with separate references.
In Choleretic effects section,
-Please add an introduction about Choleretic effects and then explain the results of Corydalis saxicola Bunting (CSB).
In Clinical applications section,
- In Icteric hepatitis section, please add references for the first two sentences.
- In Hyperbilirubinemia section, please add references for the first 5 lines.
In Toxicity assessment section,
- Line 3, “There are some studies on the toxicity of CSBTA…”, please add more than two refrences for this section.
Collectively, this is a comprehensive review about Corydalis saxicola Bunting (CSB); however, it needs some corrections and there are a lot of long and confusing sentences that should be revised to be more understandable.
Thank you so much for sharing your valuable review.
Author Response
Dear Editor:
I thank you and reviewers for the great efforts to help us in improving the quality of the manuscript (ijms-1926829). I have made point-by-point replies to your and reviewers’ specific criticisms, and revised the manuscript following the comments.

Reviewer 2 Report
After the revision of the manuscript entitled “Corydalis saxicola Bunting: A review of its traditional uses, phytochemistry, pharmacology, and clinical applications”, my comments are as follows:
1. In the part of the traditional use: Is this plant used only in the traditional medicine of China? I think their traditional use is expanding in other countries. I suggest to do a thorough search in order to record other Tharapeutical uses in other countries.
2. In the phytochemistry part: I do not think that the extracts of this plant are only alkaloids. There are other major molecules such as polyphenols, flavonoids, terpenes…. You have to look in this part. I consider it incomplete. To this end, I recommend to complete this part
3. In the pharmacological part of CSB: The phytochemical compounds responsible for the pharmacological effect should not be indicated for each activity. I recommend you to shave all these phyto chemical data in a single part which is the phyto chemical part to come to conclusions on the phyto chemical profile of this plant. The general idea on the phytochemical profail of a plant is very interesting.
4. Some abbreviations, it is necessary to indicate their meaning. Revise it's in all the maniscrit
Author Response

(The authors gave the same response as above.)

Round 2
Reviewer 1 Report
Dear authors,
In my opinion, your manuscript is publishable as it stands.
Reviewer 2 Report
the manuscript has been improved by the authors according to the recommendations given in previous revisions